# Transcatheter Treatment of Mitral Regurgitation

**DOI:** 10.3390/jcm11102921

**Published:** 2022-05-22

**Authors:** Angela McInerney, Luis Marroquin-Donday, Gabriela Tirado-Conte, Breda Hennessey, Carolina Espejo, Eduardo Pozo, Alberto de Agustín, Nieves Gonzalo, Pablo Salinas, Iván Núñez-Gil, Antonio Fernández-Ortiz, Hernan Mejía-Rentería, Fernando Macaya, Javier Escaned, Luis Nombela-Franco, Pilar Jiménez-Quevedo

**Affiliations:** Unit of Interventional Cardiology, Department of Cardiology, Cardiovascular Institute, Hospital Clínico San Carlos, Instituto de Investigación Sanitaria del Hospital Clínico San Carlos (IdISSC), Calle del Prof Martín Lagos, s/n, 28084 Madrid, Spain; angela_mcinerney@hotmail.com (A.M.); luismarroquin92@gmail.com (L.M.-D.); gabrielatirado@gmail.com (G.T.-C.); breda_82@hotmail.com (B.H.); carolina.espejo.paeres@gmail.com (C.E.); eduardopozoosinalde@yahoo.es (E.P.); albertutor@hotmail.com (A.d.A.); nieves_gonzalo@yahoo.es (N.G.); salinas.pablo@gmail.com (P.S.); ibnsky@yahoo.es (I.N.-G.); antonio.fernandezortiz@salud.madrid.org (A.F.-O.); hmejiarenteria@gmail.com (H.M.-R.); fernando.macaya.ten@gmail.com (F.M.); escaned@secardiologia.es (J.E.); luisnombela@yahoo.com (L.N.-F.)

**Keywords:** mitral valve, transcatheter, mitral valve repair, mitral valve replacement

## Abstract

Mitral valve disease, and in particular mitral regurgitation, is a common clinical entity. Until recently, surgical repair and replacement were the only therapeutic options available, leaving many patients untreated mostly due to excessive surgical risk. Over the last number of years, huge strides have been made regarding percutaneous, catheter-based solutions for mitral valve disease. Transcatheter repair procedures have most commonly been used, and in recent years there has been exponential growth in the number of devices available for transcatheter mitral valve replacement. Furthermore, the evolution of these devices has resulted in both smaller delivery systems and a shift towards transeptal access, negating the need for surgical incisions. In line with these advancements, and clinical trials demonstrating promising outcomes in carefully selected cases, recent guidelines have strengthened their recommendations for these devices. It is appropriate, therefore, to now review the current transcatheter repair and replacement devices available and the evidence for their use.

## 1. Introduction

Although the prevalence varies across countries, native mitral valve (MV) disease is the second most common valvular heart disease (VHD) in Europe [1,2]. Mitral regurgitation (MR) remains the most common disease of the MV with primary, degenerative MR, due to dysfunction of any of the components of the MV, being more common than secondary or functional MR caused by changes in the geometry of the left ventricle or left atrium. For both aetiologies, surgical repair and replacement remain the preferred treatment as per current guidelines [3,4]. However, many patients are unsuitable for surgical interventions, and as such transcatheter mitral intervention, including both repair and replacement techniques, have significantly increased in frequency. In the United States alone, the STS TVT registry reports an almost 10-fold increase in the number of transcatheter edge-to-edge repair procedures (TEER) and a 13-fold increase in the number of transcatheter MV replacement (TMVR) procedures between 2014 and 2020 [5]. This exponential increase has been facilitated by improved outcomes stemming from better patient selection, operator experience, and technical engineering advances. As a result, both American and European valvular heart disease guidelines have strengthened their recommendation for transcatheter therapies, specifically transcatheter edge-to-edge repair for both primary and secondary MR [3,4]. Experience with transcatheter replacement systems is increasing but much work remains to be done. The high variability and complex anatomy of the MV means that many patients remain unsuitable for transcatheter repair and replacement techniques. The purpose of this review is to examine the current transcatheter devices available for both repair and replacement of the mitral valve and consider potential future directions of these technologies.

## 2. Anatomical and Imaging Considerations

The mitral valve is a complex anatomical structure whose function depends on an interplay between the left atrium, the valve leaflets, the subvalvular apparatus including chordae tendineae, papillary muscles, and the left ventricle. Figure 1 outlines the anatomy of the MV on transesophageal echocardiography (TEE). The mitral valve annulus has a three dimensional “saddle shape” being higher along the anterior annulus and lower at the commissures [6,7,8]. The anterior annulus has two fibrous structures called trigones. The anterolateral trigone is contiguous with the left coronary cusp of the aortic valve while the posteromedial trigone is contiguous with the membranous septum, tricuspid annulus, and noncoronary cusp of the aortic valve. This fibrous intertrigonal region acts to reinforce the anterior annulus, while the posterior annulus is made up of myocardium and is prone to dilatation in this area. The anterior and posterior MV leaflets are anchored at the annulus with the anterior leaflet occupying one third of the annular perimeter and the remaining two thirds occupied by the posterior leaflet. The posterior leaflet is divided into three scallops (P1, P2, and P3) and although the anterior leaflet does not have the same scallop formation, it too is divided for descriptive purposes into A1, A2, and A3. The free tips of each leaflet are attached to the chordae tendineae, which in turn are attached to the papillary muscles. The chordae tendineae are classified based on their point of attachment to the leaflet: primary chordae attached to the free edge of the leaflet, secondary chordae to the ventricular surface of the leaflet, and tertiary chordae, only found at the base of the posterior leaflet, insert into the left ventricular wall. The papillary muscles are named according to their anatomical location: posteromedial and anterolateral. Dysfunction of any part of the valvular apparatus can result in mitral valve dysfunction.

Complete imaging assessment of the MV is fundamental to plan and perform transcatheter repair and replacement procedures. Imaging assessment should focus on defining the severity and aetiology of the MV disease and a full morphological assessment of the valve, annulus, and subvalvular structures, left ventricular function and the spatial relationship between the MV and the coronary sinus (CS), left circumflex artery, left ventricular outflow tract, and the aortic valve. As such, echocardiography and multidectector computed tomography (MDCT) play an important role in patient selection, procedure planning, and device selection.

## 3. Echocardiography

Echocardiography is the mainstay when assessing MV disease, and current guideline recommendations are based on echo parameters [3,4]. The most recent guidelines have modified the echo definition of severe secondary MR such that it is now aligned with that of severe primary MR. In the context of assessing suitability and planning transcatheter MV procedures, both transthoracic echocardiogram (TTE) and TEE play an important role [6,7,9]. Initial assessment using echo can determine the aetiology of mitral regurgitation (primary or secondary), grade MR severity, assess for other valvular diseases, and assess ventricular function, and both 2D and 3D TEE are essential for procedure planning and device selection. Pre-procedure TEE provides information on leaflet length, motion and calcification, and the presence of leaflet anomalies (clefts, perforations etc.), as well as useful information for planning transeptal and transapical puncture sites. Intraprocedure TEE is necessary to guide the transeptal and transapical puncture, device advancement, positioning and deployment, assessing the results of the intervention including degree of residual MR, MV gradient, paravalvular leak and left ventricular outflow tract obstruction (LVOTO). Finally, TEE acts to rule out complications prior to completing the procedure such as leaflet damage, damage to the subvalvular apparatus, pericardial effusion etc. Fusion imaging whereby the TEE image is superimposed on the fluoroscopic image is becoming increasingly available and has many advantages [10].

## 4. Multidetector Row Computed Tomography (MDCT)

MDCT has become increasingly important in the pre procedural assessment of patients undergoing many mitral valve repair procedures (particularly those targeting the MV annulus) and in TMVR where its use is mandatory prior to patient acceptance for many available devices [7]. MDCT reconstruction can provide important information regarding the proximity of the MV to surrounding structures. Particularly in mitral annuloplasty procedures (both direct and indirect), the proximity of the coronary sinus and native MV annulus to the circumflex artery may highlight the potential for circumflex artery damage. Determining suitability for TMVR and selection of appropriate TMVR devices require accurate assessment of a number of MV parameters including annulus perimeter, intercommisural and anteroposterior (AP) diameter, intertrigonal distance, and the degree and distribution of mitral annular calcification. LVOTO is a particular concern with TMVR, and MDCT can be used to predict the risk of LVOTO by simulating device implantation and measuring the neo-LVOT dimensions [11]. Figure 2 depicts parameters measured on CT when planning a TMVR and demonstrates planning of a Tendyne™ procedure.

## 5. Transcatheter Mitral Valve Repair

Transcatheter mitral valve repair (TMRr) techniques can be broadly classified into three groups: devices aimed at modifying the MV annulus, devices targeting the MV leaflets, and devices targeting the subvalvular apparatus, specifically the chordae tendineae. The aim of all these therapies is to reduce MR by increasing leaflet coaptation. A summary of devices available is provided in Table 1 and Figure 3.

### Devices Aimed at Modifying the MV Annulus

Devices aimed at modifying the MV annulus, reducing its dimensions, and thereby improving leaflet coaptation are used in secondary MR with annular dilatation. Annular modification can be performed directly (Cardioband™ system (Edwards Lifesciences, Irvine, CA, USA) and the Millipede Mitral Annuloplasty System (Boston Scientific, Marlborough, MA, USA)) or indirectly via the coronary sinus (Carillon^®^ mitral contour system™ (Cardiac Dimensions, Kirkland, WA, USA) and the ARTO™ system (MVRx Inc., San Mateo, CA, USA)).

## 6. Direct Annuloplasty Transcatheter Systems

The Cardioband™ system (Edwards Lifesciences, Irvine, CA, USA) is a transfemoral, transeptal system with Conformité Européne (CE) approval delivered with a steerable catheter via a 25Fr sheath. The device is a polyester sleeve with radiopaque markers spaced at 8 mm intervals containing a contraction wire connected to an adjusting spool, which facilitates ‘cinching’ of the device to reduce the annular dimensions. The anchors are implanted along the posterior annulus beginning at the anterolateral commissure and continuing to the posteromedial commissure. Pre procedural CT and intraprocedural coronary angiography can be used to demarcate the trajectory of the circumflex artery to avoid injury. One-year outcomes on 60 patients following Cardioband™ implantation have been reported with modification of the device following the first ten patients to solve the problem of anchor disengagement [12]. Overall technical, device and procedural success were 97%, 72%, and 68%, respectively, with two in-hospital deaths and 1-year survival of 87%. Prior to discharge, 12% of patients continued to have severe MR, 22% moderate, and 65% mild MR. Increase in MR of at least one grade was seen in 22% at 12 months. Currently, the mitral Cardioband program has been put on hold.

The Millipede IRIS Mitral Annuloplasty System™ (Boston Scientific, Massachuetts, USA) is a transfemoral, transeptal system delivered through a 24 Fr deflectable catheter. The device is a semi-rigid complete nitinol ring with a zig-zag configuration and 8 helical stainless-steel anchors at its base. The anchors attach directly to the annulus and rotate independently such that they can be ‘unscrewed’ and replaced if initial placement is not satisfactory. Once the anchors are placed, tension is applied to draw the anchors together and modify the annulus. Rogers et al. reported the outcomes in seven patients implanted with the Millipede IRIS™ with no device related death, stroke, or myocardial infarction (MI) and ≤1+ MR at 30 days in all patients [13]. The device is not yet CE or FDA approved, and further studies are awaited (NCT04147884).

## 7. Indirect Annuloplasty Transcatheter Systems

Carillon^®^ mitral contour system™ (Cardiac Dimensions, Kirkland, WA, USA) utilizes the coronary sinus (CS) to modify the mitral valve annulus accessed through the internal jugular vein using a 10 Fr system. A sizing catheter determines the length of the CS and device length required (60 or 80 mm). The device consists of two nitinol anchors: one placed distally in the great cardiac vein and the other placed proximally in the coronary sinus with a nitinol connector ribbon. Deployment of the anchors is followed by cinching, which modifies the annulus.

Initial observational studies demonstrated a good effect on MR reduction; however, many patients were unsuitable for device implantation due to risk of coronary compromise or insufficient MR reduction [14,15]. Following device modification, the TITAN II study included 36 patients, 6 of whom were not implanted due to risk of coronary compromise [16]. Of the remaining patients, 30-day MACE was 2.8%, and 1-year mortality was 23% (none of which were device related).

The REDUCE FMR trial was a blinded randomized sham-controlled trial including 120 patients randomized on a 3:1 basis to either the Carillon device (*n* = 87) or sham control (*n* = 33) [17]. The device was implanted in 84% of patients assigned to that arm (73 of 87). Two deaths (2.3%) and three MIs (3.5%) occurred in the device group in the first 30 days. Reduction in MR volume at 12 months was greater in the device group (−7.1 mL/beat in the device group versus +3.3 mL/beat in the sham control group, *p* = 0.049) as was improvement in 6-minute walk test distance and NYHA class. Lipiecki et al. reported long term results from a single centre with 6-year Kaplan–Meier estimate for mortality of 40% [18]. Better baseline function, higher LVEF and greater improvement in MR (specifically EROA) were associated with long term survival.

The ARTO™ system (MVRx Inc., San Mateo, CA, USA) modifies the MV annulus using a bridge suture between a T-bar shaped anchor placed in the great cardiac vein (GCV) and a second anchor in the interatrial septum [19]. Deployment of the ARTO system requires both access to the CS and the left atrium (via a transeptal puncture). Both delivery systems are 12 Fr. Magnetic catheters are placed both in the GCV and the left atrium and approximation of the magnetic components facilitates the passage of a wire from the catheter in the GCV to that in the left atrium, which is used to railroad the device. The anchor in the interatrial septum is akin to a PFO closure device and closes the atrial septostomy. Tightening of the device achieves the required reduction in annular dimensions. The 1- and 2-year outcomes of the MAVERIC trial have recently been reported [20,21]. Forty-five patients were included and although 7 (15.5%) had procedural complications, none were device related. Technical success was therefore 100%. A significant reduction in EROA from baseline to 1 year was noted (26.9 mm^2^ to 17.6 mm^2^, *p* = 0.0034) and MR grade was ≤2+ in 85.7% and 90.5% at 1 and 12 months, respectively.

### 7.1. Devices Targeting the MV Leaflets

Transcatheter edge-to-edge repair (TEER) devices are the most commonly used transcatheter mitral interventional devices [5] and employ a technique akin to the Alferi stitch [22] by clasping the anterior and posterior leaflets causing coaptation and creating a double orifice. Two devices employ this technique. The MitraClip™ device (Abbott, Abbott Park, IL, USA) and the PASCAL™ device (Edwards Lifesciences, Irvine, CA, USA) both of which are licensed for use in degenerative and functional MR. Recent changes to both the American and European guidelines have resulted in strong recommendations for the use of TEER techniques for the treatment of both primary and secondary MR (Table 2)

The MitraClip™ (Abbott Vascular, Abbott Park, IL, USA ) is a transcatheter, transeptal mitral valve repair system (TMVr). The current G4 iteration differs from its predecessors by having an expanded repertoire of device sizes (NT, NTW, XT, XTW) and independent leaflet grasping to allow optimization. The system consists of a 24 Fr steerable guide catheter and a clip delivery system. After transeptal crossing, the steerable catheter and clip delivery system allow orientation of the clip above and perpendicular to the leaflets. The opened clip is then passed into the ventricle and the leaflets are grasped as the clip is slowly pulled back towards the atrium. Once appropriate reduction in MR is confirmed without significant stenosis or interference with the other valvular structures, the device is released.

The MitraClip™ device is the most widely studied of the TEER devices. The EVEREST I trial demonstrated device feasibility and safety with sustained reduction in MR to ≤2+ at 6 months [23]. EVEREST II randomized patients with 3+ or 4+ MR to either TEER with MitraClip™ or surgical repair on a 2:1 basis [24]. The results showed reduced efficacy with the TEER system compared to surgical MV repair mainly driven by the greater need for surgical intervention and greater residual MR in the TEER group. As such, TEER has been mainly reserved for high-risk or inoperable patients. Echocardiographic inclusion and exclusion criteria used in the EVEREST II study have become widely accepted and used in patient selection for MitraClip procedures [25] (Table 3).

Controversy regarding this device arose with the simultaneous publication of two randomized controlled trials; COAPT [26] and MITRA-FR [27] both examining the use of MitraClip™ in secondary MR. The COAPT trial randomized 614 patients with heart failure and severe MR who remained symptomatic despite optimal guideline directed therapy to either MitraClip™ or usual care. Those in the device group had a significant reduction in annualized hospitalization rate for heart failure (HR 0.53, 95% CI 0.40–0.70, *p* < 0.001) and reduced all-cause mortality at 24 months (HR 0.62, 95% CI 0.46–0.82, *p* < 0.001. These promising results seemed at odds with those of MITRA-FR which randomized 304 patients with secondary MR to medical therapy or MitraClip™. At 12 months, no difference was seen in all-cause mortality (HR 1.11, 95% CI 0.69–1.77) or heart failure hospitalization (HR 1.13, 95% CI 0.81–1.56). Further analysis of these two trials demonstrated a number of differences in the included cohorts [28]. Specifically, those included in the MITRA-FR trial had greater LV dilation and a lesser degree of MR due to use of different guideline defined cut-offs between the two studies. Sustained positive results at 3-year follow up in the COAPT trial [29] suggest that patient selection is an important determinant of outcome following MitraClip™, and more recent guidelines are congruent with this finding [3]. Table 3 outlines the typical COAPT-like parameters suggested by guidelines.

It is important to note that all of these aforementioned studies were performed with older versions of the MitraClip™, and the current iteration (G4) has a number of improvements including an expanded range of sizes, continuous LA pressure monitoring, and individual leaflet grasping for optimizing results. A study including 59 patients treated with the G4 MitraClip™ demonstrated reduction in MR to ≤2+ occurred in 96.6% during the procedure and was sustained at 30-day follow up [30].

The PASCAL™ (Edwards Lifesciences, Irvine, CA, USA) repair system also employs a mitral valve edge-to-edge repair technique. Two systems are available, the PASCAL™ and the PASCAL Ace™, and they differ in device width (PASCAL 10 mm, PASCAL Ace 6 mm). The system consists of a steerable sheath and implant catheter and offers increased range of movement to facilitate device placement in difficult anatomies. The system has three main components: the paddles and clasps between which the mitral valve leaflets are grasped, and a central spacer which acts to reduce the EROA without applying excessive leaflet tension. Independent leaflet clasping is also a feature of these devices.

The prospective, single arm CLASP study enrolled 124 patients with both functional (69%) and degenerative (31%) severe MR. Successful implantation was achieved in 96% of patients. At 2 years, the major adverse event rate was 16.9% with 80.3% survival [31]. Thirty-day TTE assessment demonstrated MR ≤ 2+ in 97%, which was sustained at 2 years (*n* = 36) as was transvalvular gradient.

Current guidelines do not specify a preferred TEER device for TMVr and there are currently no randomized studies comparing the two. However, the CLASP IID/IIF randomized controlled trial comparing both devices is currently recruiting NCT03706833.

### 7.2. Devices Targeting the Subvalvular Apparatus

A number of devices target the subvalvular apparatus and aim to restore the tethering function of the chordae tendineae.

The NeoChord™ (Neochord Inc., St. Paul, MN, USA) artificial chordae delivery system uses an expanded polytetrafluoroethylene (ePTFE) suture to create a new chordae tendineae. It has CE approval for the treatment of MR due to prolapse or flail leaflet. The device consists of a handheld delivery device, a cartridge into which the suture is loaded, a needle, and a leaflet capture verification monitor. The procedure is transapical, requiring a lateral mini-thoracotomy. TEE guides the incision site in the LV apex to attain proper alignment with the mitral valve, and the instrument is introduced. Leaflet grasping is confirmed on the verification device, and the suture is deployed. Several sutures can be inserted, each one individually secured to a pledget on the epicardial surface of the heart.

The TACT study examined the feasibility, safety, and efficacy of the NeoChord system. From 30 patients with severe MR due to MV prolapse (Carpentier type II), 86.7% of patients had a reduction in MR to ≤2+ [32]. Major adverse events occurred in 8 patients (26.7%) within 30 days, including one death. A larger multicentre series of 213 patients was published by Colli et al. with procedure success in 96.7% [33]. Leaflet rupture with severe MR occurred in four patients, while significant bleeding occurred in eight patients (3.7%). Less than moderate MR was present in 98.5% of patients at discharge, 93% at 6 months, and 92.1% at 1 year. Overall survival at 1 year was 98%.

The HARPOON™ beating heart mitral valve repair system (Edwards Lifesciences, Irvine, CA) is a chordal system using an ePTFE suture delivered transapically via a 14 Fr introductory system. The device is positioned beneath the posterior MV leaflet, and the needle passes through the target scallop, creating a knot on the atrial surface. The suture is then withdrawn through the device, tightened, and fixed to a pledget on the epicardial surface of the ventricle. The combined results of two multicentre studies were reported by Gammie et al., including 65 patients [34]. The primary end point of successful implantation and reduction of MR to ≤moderate at 30 days was met in 91%. One patient died within 30 days. At 1-year, 98% of patients were NYHA class I-II, and 98% had ≤moderate MR.

## 8. Transcatheter Mitral Valve Replacement (TMVR)

TMVR techniques have increased in frequency in recent years as have the number of devices now available for use. TMVR continues to be reserved for inoperable or high surgical risk patients, and current guidelines have highlighted the paucity of trial data concerning their use [4]. However, TMVR can provide a solution in certain anatomical subsets that may not be amenable to percutaneous repair. In particular, mixed mitral valve disease, small MV orifice area (<3 cm^2^), broad or commissural MR jets, flail leaflet, and Barlow’s disease, all of which are unsuitable for edge-to-edge repair, may be more appropriate for TMVR [35]. Until recently, TMVR experience has mainly involved already established valve systems, used for TAVR, repurposed for use in MV pathologies. Most commonly, balloon-expanding valve systems (mainly Sapien 3™ and Sapien XT™ devices (Edwards Lifesciences, Irvine, CA) have been used to treat failed prosthesis in the MV position (valves (ViV) or annuloplasty rings (ViR)) [36,37,38,39] or in patients with severe mitral annular calcification (ViMAC) [38,40]. While ViV procedures have shown good technical and mid-term results [36,39], ViR and ViMAC procedures have been technically less successful [37,38,39,40], highlighting the complexity of the procedure and MV anatomy, which dedicated transcatheter MV replacement devices aim to overcome.

Dedicated TMVR devices, previously only delivered through the transapical route, have evolved significantly with transeptal devices now showing promise. Furthermore, greater standardization of pre-procedural imaging for planning and predicting potential complications has led to improved patient selection; however, there continues to be a large number of patients not suitable to undergo these procedures. A review of 40 patients by Coisne et al. screened for potential TMVR deemed 60% to be unsuitable due to anatomical reasons and risk of LVOTO [41]. This highlights the engineering challenges in designing dedicated MV transcatheter systems.

LVOTO remains one of the biggest concerns with TMVR. Obstruction of the “neo-LVOT” can be from prosthesis protrusion into the LVOT or from displacement of the anterior MV leaflet. Valve-in-MAC procedures are at particular risk. An estimated neo-LVOT area of ≤1.7 cm^2^ measured using a device-specific virtual valve implantation on MDCT predicts LVOT obstruction with a high sensitivity and specificity [11]. A number of solutions to decrease the risk of LVOTO have been proposed but require further investigation, including alcohol septal ablation to increase the LVOT area [42] or intentional laceration of the anterior MV leaflet (LAMPOON procedure) [43]. Hybrid surgical and transcatheter procedures aimed at reducing LVOTO by debriding MAC and excising the anterior MV followed by deployment of a transcatheter heart valve have been reported and remain an option for some patients. Other challenges include the risk of device embolization due to the wide variability of annular dimensions of the MV and its dynamic nature during the cardiac cycle. Device anchoring is therefore an important component of TMVR, with some devices having active anchoring systems and others utilizing oversizing or other anchoring mechanisms. A summary of each device is outlined in Table 4 and Figure 4. Procedural results are depicted in Figure 5. Planned studies as registered on https://clinicaltrials.gov/ (accessed on 1 December 2021) are outlined in Table 5.

## 9. Transapical TMVR Systems

Tendyne™ (Abbott Vascular, Abbott Park, IL, USA) is a transapical system consisting of two stents. The inner nitinol stent houses the valve leaflets (three leaflets of bovine pericardium) while the outer stent is D-shaped to conform to the native MV. The straight part of the D-shape is orientated towards the aortomitral continuity. The stents are connected with a polyethylene terephthalate (PET) fabric cuff that provides sealing and prevents perivalvular leak. Valve deployment begins in the left atrium and is guided by TEE. The valve is fully repositionable and retrievable during the procedure to optimize the results. Tethering to the LV is via a braided fibre secured to a pad on the epicardial surface of the ventricle.

Two-year data has been published for patients with both primary and secondary MR [44,45]. Technical success was 96% with no intraprocedural mortality; however, BARC ≥ 2 bleeding occurred in 18% of patients. Acute results showed reduced MR to ≤trace in 99% of patients and absent MR in 95.3% at 6-months and 98.4% at 1-year (61 out of 62 patients). Thirty-day, 1- and 2-year mortality was 6%, 26%, and 39%, respectively. At 2 years, 93.2% of surviving patients had no MR. A CT analysis at 1 month following Tendyne implantation suggests positive ventricular remodelling in the majority of patients [46].

Intrepid™ (Medtronic Inc., Minneapolis, MN, USA) is now available as both a transapical and transeptal system. It has a two-stent design. The inner stent houses the trileaflet bovine pericardial valve and the outer stent forms a fixation ring. The valve itself measures 27 mm and the fixation ring is available in variable sizes. Fixation and sealing are achieved in multiple ways: oversizing (10–30%) to provide radial force, conformation of the flexible atrial portion of the fixation ring to the MV annulus, the “champagne cork” (narrow neck and wider body) formation between the atrial and ventricular portions, which avoids apical displacement during systole, and three rows of friction elements that further aid fixation. The two-stent design ensures that the inner stent housing the valve leaflets maintains the same circular conformation during both systole and diastole.

The results of 50 patients of high or extremely high surgical risk patients treated using the transapical device were reported in 2018 [47]. One patient did not undergo valve implantation due to access site bleeding. Successful implantation occurred in 98% of the remaining patients. Seven died within 30 days, three of these related to the apical access site. At 3 days, all surviving patients had absent MR and no LV obstruction. In an early feasibility study of the Intrepid transfemoral, transeptal device, 14 of 15 patients had successful device implantation with ≤mild MR and no LVOTO [48].

Tiara™ (Neovasc, Minnesota, USA) is a transapical self-expanding nitinol valve with three bovine pericardial leaflets. The valve has a D-shaped configuration designed to fit MV anatomy. Ventricular anchoring structures secure the valve onto the fibrous trigones and the posterior shelf of the MV annulus. Delivery is via a 32 Fr sheath through the apex into the left atrium. The atrial portion of the device is unsheathed, orientated, and retracted into position. The ventricular portion is then released, and the valve is anchored. Plans to develop a transfemoral/transeptal deliver system were recently put on hold. An early feasibility study (TIARA-I) and extended clinical study (TIARA-II) are currently underway, and pooled preliminary results of 71 patients demonstrated no procedural deaths and 93% successful deployment [49]. Thirty-day mortality was 11.3% (8 patients, 6 of whom were cardiovascular deaths). Thirty-three patients had 30-day TTE data available and of these, 96.9% had ≤mild MR.

## 10. Transeptal TMVR Systems

The EVOQUE™ (Edwards Lifescience, Irvine, CA, USA) TMVR is a transeptal system consisting of a self-expanding nitinol frame with a bovine pericardial valve delivered via a 28 Fr sheath. The delivery capsule is aligned using TEE and passed through the mitral valve where unsheathing begins in the ventricle. As the valve is unsheathed, it is anchored by capturing the valve leaflets and subvalvular apparatus. The atrial portion of the valve has a sealing skirt to prevent PVL. Initial results for fourteen patients demonstrated technical success in 92.9% (13 of 14 patients) [50]. One required conversion to surgery for severe PVL. All-cause mortality at 30 days was 7.1%. One device related ischaemic stroke, and one procedure related disabling hemorrhagic stroke occurred. Thirty-day TTE assessment in 12 patients demonstrated ≤mild MR in 100%. Use of the EVOQUE device in tricuspid regurgitation is under investigation.

The HighLife™ TMVR (HIGHLIFE SAS, Paris, France) system consists of two components: a 31 mm ring-shaped subannular implant (SAI) that encircles the subvalvular apparatus and is delivered through an 18 Fr catheter retrogradely via the femoral artery and aortic valve, and secondly the valve component, delivered transeptally through a 39 Fr catheter that sits within the subvalvular ring. The SAI is made of nitinol, covered with PET while the valve is self-expanding nitinol with bovine pericardium. Of 30 patients implanted with HighLife technical success was reported in 27 (90%) [51]. Four major bleeding events and one conversion to open surgery occurred. Three deaths occurred within 30 days. Thirty-day TTE in the remainder showed ≤mild MR in all patients. A new iteration of the HighLife™ valve (HighLife Clarity™) with improved shape of the ventricular portion to prevent LVOTO is in development, with plans for a second valve size to be introduced.

The Sapien M3™ (Edwards Lifescience, Irvine, CA, USA) device utilizes a Sapien 3 (balloon expandable, bovine pericardium leaflets) valve and adds a subvalvular PET nitinol ‘dock’ that encircles the chordae tendineae and acts as an anchor for the prosthetic valve when it is deployed. The system is delivered transeptally (unlike the HighLife valve) via a 20 Fr system. The valve component of the system is identical to the Sapien 3 29 mm valve. The subvalvular ‘dock’ can be recaptured if necessary. The results of ten patients were reported by Webb et al. [52]. Successful implantation was achieved in 90%; one patient did not receive the implant due to a pericardial effusion requiring pericardiocentesis during dock deployment. There were no deaths or strokes at 30 days, and MR was reduced to ≤mild in 8 patients (89% of implanted patients). One patient had severe paravalvular MR, which was closed percutaneously.

The Cardiovalve™ (Cardiovalve Israel) transeptal system consists of two frames: atrial and ventricular. The valve has bovine pericardial leaflets, and the frame is designed to mirror modern day surgical mitral valve prosthesis with a low profile on both the atrial and ventricular portions. The 30 Fr system is delivered transeptally, and implantation occurs in three steps. On crossing the valve, the leaflets are grasped, and once adequate grasping and co-axial alignment has been confirmed, the atrial flange (the sealing portion of the device) is released and finally the ventricular frame is released. Three valve sizes are available. Results from five patients demonstrated 100% successful implantation with no residual MR. At 30-days, three deaths occurred (60%) with two related to bleeding events [53].

The Cephea™ TMVI system (Abbott Vascular, Abbott Park, IL, USA) is a self-expanding nitinol valve with a double disc structure and bovine pericardial leaflets. The atrial disc is positioned at the floor of the left atrium and a central column provides support and houses the pericardial leaflets. The ventricular disc then anchors to the subvalvular apparatus. One size (36 mm) is currently available. Outcomes for three patients with primary degenerative MR have been reported [54]. Successful implantation occurred in all patients (100%) without any procedural complications. Residual transvalvular MR was mild or none in all patients with all having mild paravalvular leak. All patients were alive at 6-month follow up, and all had trivial or no transvalvular MR, one had mild-moderate PVL, and the remaining two had mild PVL.

The AltaValve™ (4C Medical Technologies Inc., Minnesota, MN, USA) is a novel supra-annular, atrial fixation only device delivered via the transeptal route. The device is a self-expanding nitinol spherical frame available in sizes ranging from 50–95 mm to accommodate varying LA anatomies. The nitinol sphere contains the trileaflet bovine pericardial valve that is available in one size (27 mm). There is no active fixation of the valve, and anchoring is achieved by oversizing of the LA sphere (by 10–30%) and the annular portion of the valve (by 5–20%). The upper portion of the sphere has a large open cell structure. A PET skirt on the lower portion of the valve allows rapid endothelialization and prevents PVL. The transeptal system requires a 32 Fr transeptal sheath. A small number of cases using the AltaValve™ transapically have been reported, and one case of transeptal deployment was reported in 2019 with successful implantation and no significant residual MR [55]. Successful implantation via the transapical route has been reported by Ferreira-Neto et al. [56] with no residual MR, no LVOTO, and no events out to 30 days.

## 11. Future Directions

The prevalence of mitral regurgitation, particularly due to degenerative MV disease, is set to continue to increase in parallel with our ageing population. Transcatheter mitral valve therapies are therefore likely to expand. A number of engineering challenges, particularly in the TMVR sphere, must be overcome, including smaller delivery systems, improved sealing mechanisms to avoid paravalvular leak, and reduced device and AMVL encroachment in the LVOT to minimize the risk of LVOTO. Ancillary technologies and procedures aimed at preventing LVOTO such as septal reduction and therapies directed at the AMVL are also likely to increase in frequency, and operators will need to become proficient in these to provide reproducible results for patients. ViMAC and ViV present a number of challenges, and the results of currently recruiting and planned studies will inform much of the treatment of these entities going forward. While at present transcatheter MV therapies for the management of MR remain reserved for those patients unable to undergo surgical procedures, much like the evolution of transcatheter aortic valve replacement, it is probable that with advanced device iterations many of the current technical difficulties to the percutaneous treatment of MR will be overcome, and with appropriate clinical trial data, expansion of these therapies to lower risk groups could be on the horizon.

## 12. Conclusions

Transcatheter MV interventions have increased exponentially over the last number of years. Transcatheter repair devices, particularly TEER devices, have heretofore been most commonly used, and with appropriate patient selection, have good outcomes. However, not all patients are suitable for repair procedures, and TMVR is increasingly becoming a viable option for patients with high surgical risk and prohibitive anatomy for transcatheter repair. Although data remains scarce, a large number of prospective studies are planned in this space in the coming months and years. In parallel, engineering of currently available devices continues to evolve such that many of these devices are already on their second or third iteration and have gone from transapical to transeptal delivery systems. Given the significant morbidity associated with mitral valve disease, these advances are to be welcomed and will broaden treatment options available to these patients.

## Figures and Tables

**Figure 1 jcm-11-02921-f001:**
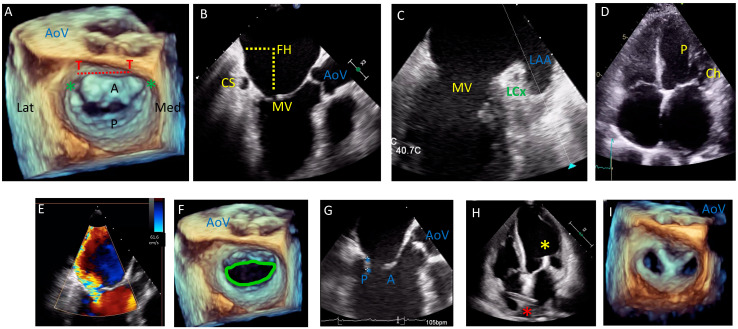
Transesophageal echocardiogram (TEE) pre-procedural assessment and intraprocedural guidance for a patient with severe secondary mitral regurgitation (MR) undergoing edge to edge repair. (**A**): mitral valve (MV) anatomy by 3D TEE showing the anterior leaflet (A) occupying one third of the annular perimeter and posterior leaflet (P) occupying two thirds. The anterior annulus has two fibrous trigones (T); the dashed red line represents the intertrigonal region. The commissures are marked with a green asterisk. The aortic valve sits anterior to the MV. Lateral (Lat) and Medial (Med) are also demarcated. (**B**,**C**) relationship between the MV and the coronary sinus (CS) and circumflex artery (LCx) on TEE; important for interventions targeting the annulus, the left atrial appendage (LAA) is seen laterally. Fossa height (FH) is demonstrated, which is important to determine feasibility of edge-to-edge repair procedures. (**D**): TTE view of the anterolateral papillary muscle (P) and its associated chordae (Ch). (**E**–**I**): pre- and intra-procedural guidance by TEE for edge-to-edge repair of the MV in a patient with secondary MR. (**E**): Severe MR with a posteriorly directed regurgitant jet (effective regurgitant orifice area (EROA) 0.81 cm^2^). (**F**): 3D assessment of MV area (4.9 cm^2^ with mean transvalvular gradient of 3 mmHg). (**G**): Posterior leaflet length (length between blue asterisk) and fossa height (not shown in this image) are important to determine feasibility of edge-to-edge repair. (**H**): Intraprocedural transthoracic echocardiogram (TTE) showing the deployed clip (yellow asterisk) with the delivery catheter traversing the interatrial septum (red asterisk). (**I**): final result on 3D TEE with a double orifice MV TEE confirmed successful implantation with no significant stenosis (MV area 2 cm^2^, mean transvalvular gradient 5 mmHg) and mild residual MR.

**Figure 2 jcm-11-02921-f002:**
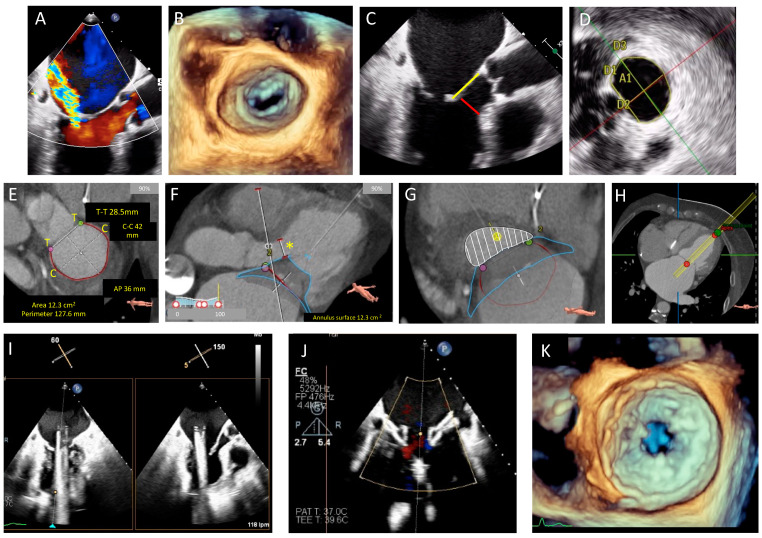
TEE and cardiac computed tomography (CT) planning for a transcatheter Tendyne™ procedure in a patient with severe MR and calcified leaflets. (**A**): Severe MR with a posteriorly directed jet, (**B**): 3D transesophageal echo (TEE) demonstrating calcified leaflet tips unsuitable for edge-to-edge repair. (**C**): Calcified mitral valve (MV) leaflet tips, anterior MV leaflet (AMVL) length 25 mm (yellow) and AMVL to septum distance of 6 mm (red) (measured to assess risk of left ventricular outflow tract obstruction (LVOTO). (**D**): 3D assessment of MV annular area (A1 = 12 cm^2^), perimeter (13.1 cm), AP dimension (D2 = 3.11 cm), inter-trigonal distance (D1 = 3 cm) and intercommisural distance (D3 = 4.6 cm). (**E**): CT planning demonstrating MV dimensions (T to T = intertrigonal distance, perimeter outlined in red, anteroposterior (AP) distance and intercommisural distance (C to C)). (**F**): Simulated Tendyne™ with predicted neo-LVOT diameter (yellow asterisk). (**G**): Simulated Tendyne with predicted neo-left ventricular outflow tract (neo-LVOT) area of 4.6 cm^2^ (white shaded area), (**H**): CT assessed apical puncture site for correct orientation with the MV. (**I**): Intraprocedural TEE guidance with X-plane views showing the delivery system across the MV and in the left atrium (LA), (**J**): Partial liberation of the Tendyne™ system (LP37M) and assessment of paravalvular leak with colour Doppler. (**K**): Liberated Tendyne™ valve with no residual MR, no paravalvular leak, mean transvalvular gradient of 5 mmHg, and no dynamic gradient in the LVOT.

**Figure 3 jcm-11-02921-f003:**
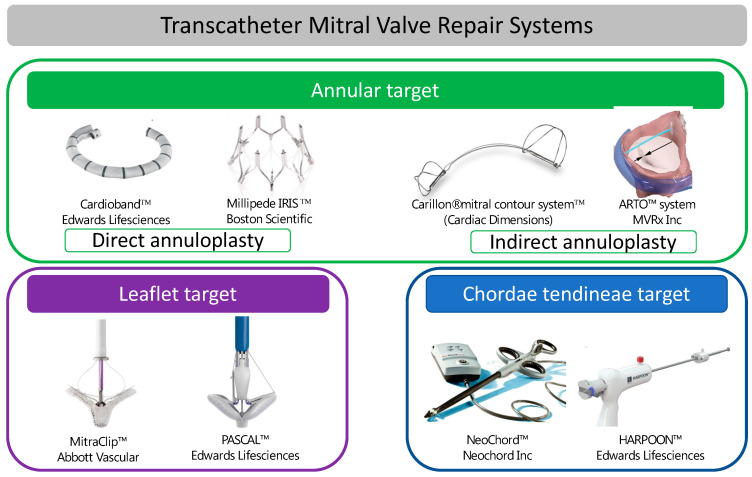
Transcatheter mitral valve repair systems.

**Figure 4 jcm-11-02921-f004:**
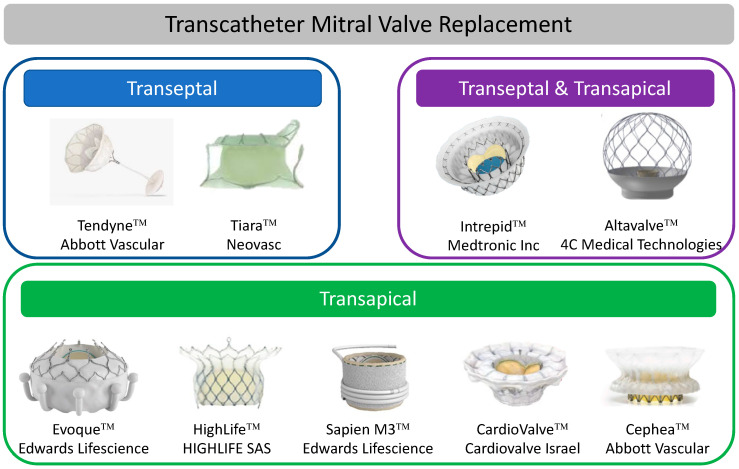
Transcatheter Mitral Valve Replacement.

**Figure 5 jcm-11-02921-f005:**
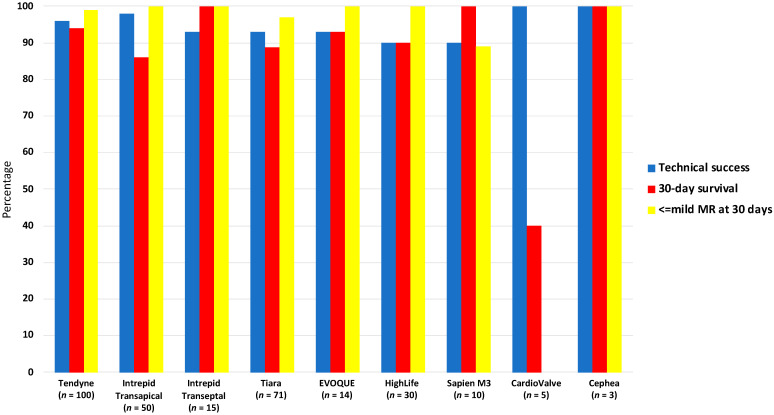
Procedural results following transcatheter mitral valve replacement.

**Table 1 jcm-11-02921-t001:** Mitral valve repair systems.

Device Target	Device	Device Description	Delivery System	Mitral Valve Access	Primary Mitral Regurgitation	Secondary Mitral Regurgitation	Approval Status
**Mitral Valve Annulus**							
Direct annuloplasty	Cardioband™	Polyester sleeve with anchors and a tightening wire to adjust annular dimensions	25 Fr sheath, steerable catheter	Transeptal	No	Yes	Conformité Européne (CE) marked
	Millipede™	Semi-rigid complete nitinol ring with 8 helical stainless-steel anchors	24 Fr deflectable catheter	Transeptal	No	Yes	-
Indirect annuloplasty	Carillon™ mitralcontour	Proximal and distal nitinol anchors connected by a nitinol connector ribbon	10 Fr	Via coronary sinus	No	Yes	CE marked
	The ARTO™ system	A bridge suture connects an anchor in the great cardiac vein (GCV) with an anchor in the interatrial septum	12 Fr	Via coronary sinus and transeptal puncture	No	Yes	-
**Mitral Valve Leaflets**							
Edge to edge repair	MitraClip™	Cobalt chromium device with a polyester coating. Consists of two arms that are opened to grasp the leaflet edges and closed to approximate the leaflets creating a figure of 8 double orifice.	24 Fr	Transeptal	Yes	Yes	CE markedFDA approved
	PASCAL™	Consists of two paddles, two clasps, and a spacer. The leaflets are grasped between the paddles and clasps, and the spacer acts to reduce the effective regurgitant orifice area (EROA)	22 Fr	Transeptal	Yes	Yes	CE markedFDA approved
**Chordae Tendineae**							
Implantation of new chordae tendineae	NeoChord™	Expanded polytetrafluoroethylene (ePTFE) suture creates a neochord that is tightened and attached to a pledget on the epicardial surface of the ventricle	-	Transapical	Yes	No	CE marked
	HAPOON™	Expanded polytetrafluoroethylene (ePTFE) chordal system	14 Fr	Transapical	Yes Only in severe MR due to PMVL prolapse	No	CE marked

**Table 2 jcm-11-02921-t002:** Table comparing current and previous guideline recommendation for transcatheter MV repair.

	ACC/AHA Guidelines		ESC Guidelines	
	2014	2020	2017	2021
**Primary ** **Mitral ** **Regurgitation** **(PMR)**	May be considered in severely symptomatic patients (New York Heart Association class (NYHA) III or IV) with prohibitive surgical risk **IIb B recommendation**	Reasonable in severely symptomatic patients (NYHA III or IV) of high or prohibitive surgical risk if anatomy is amenable **2A B-NR recommendation**	May be considered for symptomatic severe PMR with suitable anatomy and judged to be at inoperable or high surgical risk by the heart team **IIB C recommendation**	May be considered for symptomatic patients with suitable anatomy and judged to be at inoperable or high surgical risk by the heart team **IIB B recommendation**
**Secondary Mitral ** **Regurgitation** **(SMR)**	No recommendation	Reasonable in patients with persistent symptoms on optimal guideline directed medical therapy (GDMT) and who have appropriate anatomy and with LVEF 20–50%, LVESD ≤ 70 mm and PASP ≤ 70 mmHg **2a B-R recommendation**	May be considered when revascularization is not indicated and surgical risk is not low with LVEF > 30% who remain symptomatic despite optimal GDMT (including cardiac resynchronizing therapy (CRT)) **IIb C recommendation**	Should be considered in selected patients not eligible for surgery and fulfilling criteria suggesting an increased chance of responding to treatment **IIa B recommendation**
			May be considered in patients with LVEF < 30% with no revascularization option and remain symptomatic on optimum GDMT (including CRT) after careful evaluation for ventricular assist device or heart transplant **IIb C recommendation**	May be considered by the heart team in high-risk symptomatic patients not fulfilling criteria suggesting an increased chance of responding to TEER after careful evaluation for ventricular assist device or heart transplant **IIb C recommendation**

**Table 3 jcm-11-02921-t003:** EVEREST II and COAPT clinical and echocardiography criteria for MitraClip™ device.

**EVEREST Echocardiographic Criteria**
Mitral valve orifice area > 4.0 cm^2^
Transvalvular gradient < 4 mmHg
Width of flail segment < 15 mm
Flail gap < 10 mm
Coaptation depth < 11 mm
Mobile leaflet length > 10 mm
**COAPT Criteria**
Secondary mitral regurgitation
At least one HF admission in the previous year or increased natriuretic peptide
NYHA ≥ II
Left ventricular ejection fraction (LVEF) 20–50%
Left ventricular end-systolic diameter ≤ 70 mm

**Table 4 jcm-11-02921-t004:** Mitral valve replacement systems.

Device	Device Description	Delivery System	Recapture	Valve Sizes	Mitral Valve Access	Primary Mitral Regurgitation	Secondary Mitral Regurgitation	Approval Status
**Transapical TMVR systems**								
Tendyne™	A two-stent configuration: Inner stent housing the bovine pericardial valve. An outer stent that conforms to the native MV. Stents are connected with a PET fabric cuff. The valve is tethered to the LV and connected to an epicardial pad to hold it in position	34 or 36 Fr depending on valve size	Repositionable and recapturable	Valve has two profiles Low profile: EOA 2.2 cm^2^ Standard profile: EOA 3.0 cm^2^ Outer stent available in a number of sizes to conform to mitral annulus anatomy	Transapical	Yes	Yes	CE marked
Tiara™	Self-expanding nitinol stent with a trileaflet bovine pericardial valve. D-shaped configuration to conform to the MV annulus	32 for 35 mm valve 36 Fr for 40 mm valve	Repositionable and recapturable	Two sizes: 35mm and 40 mm	Transapical	Yes	Yes	-
**Transapical and Transeptal systems**								
Intrepid™	A two-stent configuration: Inner stent housing the bovine pericardial valve. Outer stent forms a fixation ring. Both stents are covered with PET	35 Fr transapical 35 Fr transseptal	Repositionable and recapturable	Single size valve: 27 mm Outer stent available in two sizes (42 and 48 mm)	Transapical Transseptal	Yes	Yes	-
AltaValve™	Self-expanding supra-annular nitinol sphere housing a 27 mm bovine pericardial valve.	32 Fr	Recapturable and retrievable	27 mm valve Spherical frame ranging from 50–90 mm	Transapical Transeptal	Yes	Yes	-
**Transeptal TMVR systems**								
EVOQUE™	Self-expanding nitinol frame with a bovine pericardial valve. The valve is anchored by capturing the native MV leaflets and subvalvular apparatus. An atrial sealing skirt prevents PVL	28 Fr	No	44 and 48 mm	Transeptal	Yes	Yes	-
HighLife™	Two components consisting of a subannular ring implant delivered retrogradely via the femoral artery and aortic valve, and the valve component delivered transeptally	30 Fr for transeptal valve delivery 18Fr femoral artery access for subannular ring implant	No	28 mm valve and ring	Transeptal	Yes	Yes	-
Sapien M3™	Two components consisting of a subvalvular “dock”, which encases the balloon expandable Sapien valve (29 mm)	20 Fr	Subvalvular “dock” component is recapturable	29 mm valve	Transeptal	Yes	Yes	-
CardioValve™	Two nitinol self-expanding frames: atrial and ventricular encasing bovine pericardial leaflets.	30 Fr	-	Three sizes available covering commissural diameters from 36 to 53 mm	Transeptal	Yes	Yes	-
Cephea™	Double disc system connected via a central column that houses the bovine pericardial valve.	Not stated	Recapturable	One size available with a 36 mm central waist	Transeptal	Yes	Yes	-

**Table 5 jcm-11-02921-t005:** Upcoming studies in transcatheter mitral valve replacement (TMVR) technologies.

Device	Study Name	Study Design	Inclusion Criteria	Primary Outcome
Tendyne	Clinical trial to evaluate the safety and effectiveness of using the Tendyne mitral valve system for the treatment of symptomatic mitral regurgitation (MR) (SUMMIT) NCT03433274	Prospective multicentre study with three cohorts: (a) Randomized cohort: 1:1 basis with MitraClip (b) Non-randomized cohort © MAC cohort Number of participants: 958	Symptomatic, moderate-severe, or severe MR, or severe mitral annular calcification (MAC)	(a) Randomized cohort: Survival free of HF hospitalization at 12 months (b) Non-randomized cohort: composite of all-cause mortality, CV related rehospitalization, stroke, or MV reintervention or reoperati©(c) MAC cohort: survival free of HF rehospitalization at 12 months
Intrepid	Transcatheter mitral valve replacement with the medtronic Intrepid™ TMVR system in patients with severe symptomatic MR (APOLLO) NCT03242642	Multicentre, single arm, non-randomized study with two cohorts (a) Primary cohort: moderate-severe or severe MR not suitable for TEER or surgical MVR (b) MAC cohort: moderate-severe or severe-MR with MAC	Moderate-severe or severe MR High surgical risk Not suitable for TEER	(a) Primary cohort: all-cause mortality or heart failure hospitalization at 30 days or KCCQ improvement < 10 composite (b) MAC cohort: composite of all-cause mortality and heart failure hospitalization
EVOQUE	Edwards EVOQUE EOS mitral valve replacement: investigation of safety and performance after mitral valve replacement with a transcatheter device (MISCEND) NCT02718001	Multicentre, prospective, single arm non-randomized study examining the safety and performance of the EVOQUE device in MR Number of participants: 83	Symptomatic mitral regurgitation High surgical risk Meeting anatomical criteria for the EVOQUE device	Major adverse events within 30 days
HighLife	Feasibility study of the HighLife 28mm trans-septal trans-catheter mitral valve in patients with moderate-severe or severe mitral regurgitation and at high surgical risk NCT04029363	Multicentre, single arm non-randomized study evaluating the feasibility, safety, and performance of the HighLife 28 mm TMVR Number of participants: 50	Moderate-severe or severe MR High risk for surgery Meeting anatomical criteria for the HighLife valve	Major adverse events within 30 days
HighLife Clarity	HighLife TSMVR feasibility study of the open cell CLARITY valve in patients with moderate-severe or severe MR, high surgical risk, and with a high risk for left ventricular outflow tract obstruction (LVOTO) NCT04888247	Open label, single centre, single arm, non-randomized study to assess the feasibility, safety, and performance of the HighLife CLARITY transeptal mitral valve replacement system Number of participants: 15	Moderate-severe or severe MR High surgical risk Meets anatomical criteria for HighLife Clarity valve High risk of LVOTO	Technical success: (a) Successful vascular access delivery and retrieval (b) Deployment and correct positi©ng (c) Freedom from additional emergency surgery or re-intervention related to the device or access
Sapien M3	Sapien M3 system transcatheter mitral valve replacement via transseptal access. The ENCIRCLE trial NCT04153292	Open label, single arm non-randomized study with two cohorts (a) Patients deemed unsuitable for surgical MVR (b) Patients with failed attempt at TEER Number of participants: 400	Moderate-severe or severe MR Unsuitable for surgical MVR due to clinical anatomical or technical considerations Failed attempt at TEER	Death and/or HF rehospitalization
CardioValve	CardioValve transfemoral mitral valve system (AHEAD) (United States) NCT03813524	Open label, multicentre, single arm, non-randomized study (United States centres) Number of participants: 15	Symptomatic severe MR High surgical risk LVEF ≥ 30% Cardiac index > 2.0 Patients must be receiving GDMT for at least 30 days prior to enrollment	Technical success (a) Successful access, delivery, and retrieval of the device (b) Successful deployment and correct pos©oning (c) Freedom from emergency surgery or reintervention related to the device Without procedure death, stroke, or device dysfunction
	European feasibility study of the CardioValve transfemoral mitral valve system (AHEAD study)	Open label, multicentre, single arm, non-randomized study (European centres) Number of participants: 30	Symptomatic severe MR High surgical risk Anatomy suitable for the CardioValve device Patients must be receiving GDMT for at least 30 days prior to enrolment (or CRT if indicated)	Freedom from all-cause mortality and major adverse events
AltaValve	AltaValve early feasibility study protocol	Open label, multicentre, single arm, non-randomized study Number of participants: 15	Severe symptomatic MR High surgical risk	Major adverse cardiac events at 30 days (death, stroke, and MV related repeat intervention)

## Data Availability

Not applicable.

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
