# Peer review of "Transcatheter Treatment of Mitral Regurgitation"

_jcm, 2022, doi:10.3390/jcm11102921_

Round 1

Reviewer 1 Report

I read with interest the review article on transcatheter treatment options for the mitral valve. The authors have reviewed the subject well. However, since the author's primary focus was on MR and did not include the most frequent transcatheter treatment for mitral valve namely BMV in the review, the heading may be modified accordingly.

In figure 1 authors have shown the trigones incorrectly. They are commissures, trigones are a few mm inside the commissures. In mitral valve anatomy, authors should include commissures in the description.

Similarly, the length between AML and septum was measured as 6 mm. This looks incorrect

A few words on the future trend may also be incorporated.

Hybrid TMVR option can also be discussed

Author Response

Reviewer 1

We would like to thank Reviewer 1 for their comments. We have made every effort to address these comments and provide a point-by-point explanation below. We feel that the Reviewer’s comments have greatly enhanced our manuscript.

I read with interest the review article on transcatheter treatment options for the mitral valve. The authors have reviewed the subject well. However, since the author's primary focus was on MR and did not include the most frequent transcatheter treatment for mitral valve namely BMV in the review, the heading may be modified accordingly.

Thank you for this comment. We decided to concentrate on mitral regurgitation as we understand another article regarding mitral stenosis will also be published. We agree that our title could be modified, and we have now titled the article “Transcatheter treatment of mitral regurgitation”

In figure 1 authors have shown the trigones incorrectly. They are commissures, trigones are a few mm inside the commissures. In mitral valve anatomy, authors should include commissures in the description.

Thank you for this suggestion. We have now modified figure 1 to demonstrate that the trigones are situated on the valve ring and have added an asterisk to show both commissures.

Similarly, the length between AML and septum was measured as 6 mm. This looks incorrect

We have looked again at this image. This measurement was made by our interventional echocardiographers and is 6mm. We therefore have not changed the figure legend.

A few words on the future trend may also be incorporated.

Thank you for this suggestion. We have now incorporated a paragraph on “future directions” (lines 507-523)

Hybrid TMVR option can also be discussed

Thank you for this suggestion, we have added a short note on the option of hybrid procedures (lines 367-370).

Reviewer 2 Report

The manuscript  is a review of transcatheter devices available for both repair and replacement of the mitral valve . The review is in time and very up to date, as during the last decade transcatheter mitral intervention including both repair and replacement techniques have significantly increased in frequency.

The authors analize 57 publications and present very thorough and comprehensive review of the mitral valve anatomy, investigation methods, including echocardiography and MDCT as well as very wide and deep analysis of different transcatheter devices. Though the authors present a lot of studies and trials, as well as present repair and replacement systems in the tables, it would be appreciated if the authors could include the number of pts, who received this or that system worldwide in the same tables. Another very important item is indications and contraindications for transcatheter replacement of the MV, what should be included in the review. If it is still not possible, owing to the data available, to  state exact indications and contraindications, it also should be mentioned in the review.  

The English level of the manuscript is high. The manuscript will be very interesting to the wide auditorium of medical specialists, especially cardiologists, cardiac surgeons, etc.

Author Response

Reviewer 2

The manuscript  is a review of transcatheter devices available for both repair and replacement of the mitral valve . The review is in time and very up to date, as during the last decade transcatheter mitral intervention including both repair and replacement techniques have significantly increased in frequency.

The authors analize 57 publications and present very thorough and comprehensive review of the mitral valve anatomy, investigation methods, including echocardiography and MDCT as well as very wide and deep analysis of different transcatheter devices.

We thank the Reviewer for his/her comments on our manuscript. We have considered all the points mentioned by the reviewer and have provided a point-by-point response to each below. We feel that manuscript has been greatly enhanced by the reviewers comments and thank them for their time and effort.

Though the authors present a lot of studies and trials, as well as present repair and replacement systems in the tables, it would be appreciated if the authors could include the number of pts, who received this or that system worldwide in the same tables.

We thank the Reviewer for this suggestion. We have tried to find this data for the Reviewer but have been unsuccessful. We can estimate from previous publications and from the Abbott website that approximatlet 150000 MitraClips have been implanted world wide. However, we cannot be sure of the accuracy of this data and it may be incorrect even by several thousand. For other procedure types it has been even more difficult.  Many of the company websites do not have this data and there is no way to estimate it. The only thing we can say for sure is how many patients have been included in studies and we tried to provide this data in the manuscript. However, the number included in studies and the number actually implanted worldwide (on compassionate grounds etc) may be vastly different. As such we have not included this data in the tables. If the Reviewer can direct us to this data, we would be delighted to include it.

Another very important item is indications and contraindications for transcatheter replacement of the MV, what should be included in the review. If it is still not possible, owing to the data available, to state exact indications and contraindications, it also should be mentioned in the review.  

Thank you for this comment. We have amended the document to state that TMVR remains a procedure for surgical ineligible patients. We also have suggested that TMVR may be a reasonable option in patients who have contraindications for transcatheter edge-to-edge repair such as mixed mitral valve disease, small MV orifice area, flail leaflet or non-central MR jets. Additionally in the manuscript we have mentioned that ViMAC and ViV remain difficult anatomies for treatment with TMVR and that the risk of LVOTO remains a significant factor that contraindicates this procedure in a number of patients. We hope these comments will be to acceptable to the reviewer and of course we are willing to readdress this if the Reviewer deems it appropriate. Lines 351-356, 370-373, 374-380.

The English level of the manuscript is high. The manuscript will be very interesting to the wide auditorium of medical specialists, especially cardiologists, cardiac surgeons, etc.

We thank the Reviewer for this assessment of our manuscript and hope it will be of interest to a wide audience.
